# Multiple Myeloma Relapse Is Associated with Increased NFκB Pathway Activity and Upregulation of the Pro-Survival BCL-2 Protein BFL-1

**DOI:** 10.3390/cancers13184668

**Published:** 2021-09-17

**Authors:** Ingrid Spaan, Anja van de Stolpe, Reinier A. Raymakers, Victor Peperzak

**Affiliations:** 1Center for Translational Immunology, University Medical Center Utrecht, Utrecht University, 3584 CX Utrecht, The Netherlands; i.spaan-2@umcutrecht.nl; 2Precision Diagnostics, Philips Research, 5656 AE Eindhoven, The Netherlands; anja.van.de.stolpe@philips.com; 3Department of Hematology, University Medical Center Utrecht, Utrecht University, 3584 CX Utrecht, The Netherlands; r.raymakers@umcutrecht.nl

**Keywords:** relapsed/refractory multiple myeloma, NFκB signaling pathway, signal transduction pathway activity, pro-survival BCL-2, BFL-1, BCL2A1

## Abstract

**Simple Summary:**

Multiple myeloma (MM) is a blood cancer that is still incurable because patients become insensitive to available treatments. The cancerous cells in MM misuse a signaling route, called the NFκB pathway, to make MM more difficult to treat. In our study, we used a new method to measure NFκB pathway activity in cancer cells of MM patients at different stages of disease. We found that NFκB pathway activity remains unchanged during disease development, is independent of the life expectancy of MM patients, and does not predict how well the cancer cells will respond to treatment. However, the cancer cells that survive after treatment of MM patients do show higher NFκB pathway activity. In a subgroup of these patients, the cancer cells also showed higher BFL-1 gene expression. BFL-1 can enhance cancer cell survival after treatment and may therefore be a potential new candidate to target in these patients.

**Abstract:**

Multiple myeloma (MM) is a hematological malignancy that is still considered incurable due to the development of therapy resistance and subsequent relapse of disease. MM plasma cells (PC) use NFκB signaling to stimulate cell growth and disease progression, and for protection against therapy-induced apoptosis. Amongst its diverse array of target genes, NFκB regulates the expression of pro-survival BCL-2 proteins BCL-XL, BFL-1, and BCL-2. A possible role for BFL-1 in MM is controversial, since BFL-1, encoded by *BCL2A1*, is downregulated when mature B cells differentiate into antibody-secreting PC. NFκB signaling can be activated by many factors in the bone marrow microenvironment and/or induced by genetic lesions in MM PC. We used the novel signal transduction pathway activity (STA) computational model to quantify the functional NFκB pathway output in primary MM PC from diverse patient subsets at multiple stages of disease. We found that NFκB pathway activity is not altered during disease development, is irrespective of patient prognosis, and does not predict therapy outcome. However, disease relapse after treatment resulted in increased NFκB pathway activity in surviving MM PC, which correlated with increased *BCL2A1* expression in a subset of patients. This suggests that BFL-1 upregulation, in addition to BCL-XL and BCL-2, may render MM PC resistant to therapy-induced apoptosis, and that BFL-1 targeting could provide a new approach to reduce therapy resistance in a subset of relapsed/refractory MM patients.

## 1. Introduction

Multiple myeloma (MM) is a hematological malignancy that is characterized by clonal proliferation of antibody-secreting plasma cells (PC) that typically reside in the bone marrow (BM) [1]. The clonal evolution that underlies malignant transformation in MM results in extensive intra- and interpatient heterogeneity. Gene expression profiling identified 10 distinct molecular clusters in newly diagnosed MM. These include four translocation clusters (CD-1, CD-2, MS, MF), a hyperdiploid cluster (HY), a cluster with proliferation-associated genes (PR), a cancer testis antigen-overexpressing cluster (CTA), and a cluster that is characterized by high expression of genes involved in the NFκB pathway [2]. Mutations in NFκB genes are reported to be most prevalent in MM compared to all other human malignancies [3]. Two studies by Keats et al. and Annunziata et al. demonstrated that approximately 20% of MM patients and 40% of MM cell lines harbor at least one genetic lesion in MM PC that results in increased or constitutive NFκB pathway activation [4,5]. Two more recent studies, using next-generation sequencing, confirmed recurrent mutations in MM PC that either result in downregulation or loss of function of NFκB negative regulators, or overexpression or gain of function of positive NFκB regulators [6,7]. The majority of mutations in NFκB genes exhibit a low individual frequency, indicating that NFκB pathway-activating mechanisms are heterogeneous between patients [8].

In addition to the clonal evolution of MM PC, disease progression is accompanied by the development of a permissive BM microenvironment. Various BM components physically bind to MM PC or secrete cytokines and growth factors to promote NFκB signaling. These include APRIL and BAFF, which are important survival factors for healthy and malignant PC [9,10]. In MM PC, NFκB signaling regulates cell proliferation via cell cycle regulators, immortalization via telomerase, angiogenesis via VEGF, and intrinsic apoptosis resistance via pro-survival proteins. In addition, NFκB-mediated upregulation of adhesion molecules and matrix proteases intensifies the interaction with the BM microenvironment, thereby promoting malignant transformation, disease progression, and therapy resistance [11]. Vice versa, MM PC stimulate NFκB signaling in cellular components of the BM microenvironment to promote an MM-permissive niche. This results in increased pro-survival IL-6 expression by BM stromal cells, promotion of osteolytic bone disease by osteoclasts, and facilitates immune evasion by activation of myeloid-derived suppressor cells [8].

The numerous processes that contribute to aberrant NFκB signaling in MM and the pivotal role of NFκB signaling on various aspects of disease emphasizes the need to target this pathway for MM treatment. Current therapeutic regimens generally consist of multiple cycles of triple-drug combinations that include a proteasome inhibitor (PI), a glucocorticoid, and an immune modulatory drug (IMID) or chemotherapy. This is combined with high-dose melphalan and an autologous stem cell transplant (ASCT) for eligible patients [12]. Many of the effective anti-myeloma drugs that are included in this therapeutic armamentarium affect NFκB signaling as their primary or secondary target. The frequently administered PI bortezomib and IMID thalidomide were both reported to inhibit NFκB signaling in MM [13,14]. Although these treatment regimens have improved the life expectancy of MM patients in recent last decades, the majority of patients gradually develop resistance to all available agents, relapse, and eventually become refractory to therapy [15].

Development of therapy resistance is a continuous threat in MM treatment, with sub-clonal heterogeneity that evolves during disease progression through selection of drug-resistant clones [16]. One mechanism to promote MM PC survival is by developing resistance to the intrinsic apoptosis pathway. This can be accomplished by overexpressing pro-survival BCL-2 proteins, of which the members BCL-XL (*BCL2L1*), BFL-1 (*BCL2A1*), and possibly BCL-2 (*BCL2*) are direct NFκB target genes [17]. BCL-XL and BCL-2 are known to contribute to apoptosis resistance in MM, but a role for BFL-1 in MM PC is far more controversial. Several studies have demonstrated the role of BFL-1 in the development and activation of B lymphocytes and reported BFL-1/*BCL2A1* overexpression in B cell malignancies [18]. However, the transcriptional repressor Blimp-1 that is required for the differentiation of B cells into antibody-secreting PC inhibits BFL-1/*BCL2A1* expression [19].

To gain more insight into the role of NFκB signaling and its pro-survival BCL-2 target genes in MM, we used the novel signal transduction pathway activity (STA) model that was described by Verhaegh et al. and Van de Stolpe et al. [20,21]. This Bayesian computational model infers NFκB pathway activity from measurements of mRNA encoding direct target genes of the NFκB transcription factors. This model thus allows for the quantification of functional NFκB pathway activity, irrespective of the mode of pathway activation, which could vary from increased signaling at the receptor level, intrinsic pathway mutations, or activation by cross-talk with additional signaling pathways. Using this STA model, we determined NFκB pathway activity in Affymetrix microarray datasets that include purified PC at multiple stages of disease and from diverse MM subsets, and correlated NFκB pathway activity with mRNA expression encoding pro-survival BCL-2 family members.

## 2. Materials and Methods

### 2.1. NFκB Signal Transduction Pathway Activity (STA) Analysis

The development of Bayesian network models to measure signal transduction pathway activity (STA), and the development and validation of the STA test to measure the activity of the NFκB signaling pathway have been described in detail previously [20,21]. In brief, the computational network model for signaling pathways is constructed to infer the probability that the pathway-driving transcription factor is actively transcribing mRNA of its target genes. The Bayesian network describes the causal relation where the measured intensity of microarray probesets is dependent on the activity of target gene transcription, which is, in turn, causally related to the activity of the transcription complex. These relations are probabilistic in nature. Selection of target genes of the pathway-driving transcription factors has been based on literature insights from in vitro and in vivo studies assessing if the gene promotor region contains a transcription factor response element, if the transcription factor binds to this response or enhancer element, the promotor functionality, and differential expression upon pathway activation [21]. The NFκB STA contains 50 probesets representing 29 unique target genes and was calibrated on Affymetrix HG-U133Plus2.0 data of samples with ground truth information about their pathway activity state [20].

### 2.2. Microarray Datasets

We used the Gene Expression Omnibus (GEO; https://www.ncbi.nlm.nih.gov/geo/ (accessed on 18 February 2020)) for publicly available datasets containing Affymetrix HG-U133Plus2.0 microarray data of gene expression in PC samples from specified donor/patient subsets at various disease stages. In all datasets, samples were enriched for PC by anti-CD138 immuno-magnetic bead selection, resulting in a purity of >80% PC. All samples were subjected to extensive quality control before the data were used as input for the STA model to calculate the probability of the NFκB pathway being active, as previously described [20]. Details of the datasets and included samples are summarized in Table 1.

### 2.3. Statistical Analysis

The STA pathway activity score was normalized to a score ranging from 0 to 100, which can be used in a quantitative manner to identify differences in NFκB pathway activity between samples, and visualized in violin plots. All included samples are represented by individual data points, solid red lines indicate the median, and dashed lines indicate the quartiles of the population. Statistical analysis was performed by GraphPad Prism 8. Multiple groups within one dataset were compared using one-way ANOVA with Tukey’s correction for multiple comparisons. Correlation was determined by linear regression analysis and reported as correlation coefficient (R) values. For all tests, a *p* value <0.05 was considered statistically significant.

## 3. Results

### 3.1. NFκB Pathway Activity Is Stable during MM Development and Prognosis but Is Significantly Higher in a Subgroup of MM Patients with a Molecular NFκB Signature

MM is consistently preceded by a pre-malignant phase that is referred to as monoclonal gammopathy of undetermined significance (MGUS) [26], and a subset of patients also go through a still asymptomatic phase called smoldering MM (SMM) [27]. Since NFκB signaling activation is one of the events associated with MGUS-to-MM progression [28], we quantified NFκB signaling activity during MM development. We performed NFκB STA analysis on dataset GSE5900 that contains CD138-purified samples of healthy donors, and MGUS and SMM patients. The SMM samples showed a median NFκB activity score of 29.2 (range 14.0–57.9), which did not significantly differ from the NFκB activity scores in the MGUS samples (median 31.4, range 16.2–55.5) and healthy donor PC (median 29.6, range 17.0–49.6; Figure 1A). This indicates that functional NFκB pathway activity remains stable during the early phases of disease development from healthy PC to malignant MM PC.

Symptomatic MM patients can be stratified into three risk groups using the international staging system (ISS), varying between a median survival of 62 months in stage 1 to a median survival of only 29 months in stage 3 [29]. We analyzed NFκB STA on dataset GSE19784 that contains CD138-purified samples of newly diagnosed MM patients with a specified ISS stage. Samples in the poor-prognosis ISS 3 stage showed a median NFκB activity score of 37.4 (range 18.8–60.5), which did not significantly differ from the NFκB activity scores in ISS 2 (median 35.1, range 24.4–56.2) and ISS 1 (median 36.0, range 17.0–60.4; Figure 1B). This indicates that functional NFκB pathway activity in MM PC from newly diagnosed patients is stable, irrespective of patient prognosis.

The samples of newly diagnosed MM patients from dataset GSE19784 could also be stratified based on molecular clusters, as performed by Broyl et al. The NFκB molecular cluster identified in this study was defined by a high expression of genes involved in the NFκB pathway and was reported to have significantly greater NFκB indexes, as reported by Keats et al. and Annunziata et al., compared to the other molecular clusters [2]. STA analysis of samples in the NFκB cluster showed a median NFκB activity score of 44.0 (range 36.6–60.5; Figure 1C). This was significantly higher than the NFκB activity scores in clusters HY (median 37.4, range 22.7–56.9), MS (median 33.0, range 25.9–51.5), PR (median 31.6, range 18.8–44.1), MF (median 31.4, range 17.0–48.2), and CD-1 (median 31.2, range 22.1–44.0). These data show that the high expression of NFκB-associated genes is accompanied by high functional activity of the NFκB pathway. In addition, when NFκB scores were compared between all included molecular clusters, no statistical differences could be observed, except for the increase in the NFκB cluster compared to clusters CD-1, MF, MS, HY, and PR (Appendix A). However, the NFκB activity score in the NFκB molecular cluster was not significantly higher compared to clusters CD-2 (median 37.2, range 24.4–60.4) and CTA (median 35.6, range 26.7–53.9). This further underlines the significance of measuring the functional pathway output by STA, on the target gene expression level, to allow the identification of all patient samples with high NFκB signaling activity.

### 3.2. NFκB Pathway Activity Does Not Predict Therapy Response but Is Significantly Increased after MM Relapse

We hypothesized that the heterogeneity in NFκB signaling activity that was observed in MM PC of newly diagnosed patient samples in GSE19784 could also be reflected in the heterogeneous clinical response of MM patients to treatment. We therefore performed NFκB STA analysis on dataset GSE68871 that contains CD138-purified samples of newly diagnosed MM patients which were subsequently treated with first-line VTD (bortezomib-thalidomide-dexamethasone) induction therapy. MM PC from patients with a complete response (CR) had a median NFκB activity score of 50.8 (range 27.0–79.0) before the start of therapy (Figure 2A). This was not significantly different from patients with a near-clinical response (nCR; median 57.0, range 32.8–74.0), very good partial response (VGPR; median 56.4, range 33.4–79.3), partial response (PR; median 55.5, range 27.8–82.3), or stable disease (SD; median 61.8, range 46.4–71.7). 

The observation that the NFκB pathway activity has no significant impact on response to first-line therapy was validated by analysis of a second dataset, GSE19554. This dataset includes CD138-purified longitudinal samples that were isolated from MM patients undergoing total therapy, consisting of chemotherapy-based induction therapy, ASCT, and maintenance/consolidation therapy. For all included patients, samples were taken at diagnosis, and after induction therapy prior to the first ASCT. A subgroup of patients relapsed after the first ASCT, and extra samples were taken after second-line induction therapy, and after the second ASCT before starting maintenance/consolidation therapy. STA analysis showed no significant difference in the NFκB activity score between patients undergoing one ASCT and patients requiring two ASCTs when samples were taken at diagnosis (one ASCT: median 36.0, range 27.0–61.1; two ASCTs: median 38.4, range 28.3–57.7), or after first-line induction therapy (one ASCT: median 50.8, range 35.1–57.4; two ASCTs: median 45.5, range 27.0–62.7; Figure 2B). 

For the subgroup of patients that relapsed after first-line total therapy, extra samples were taken after administration of second-line induction therapy. STA analysis of these samples showed a significant increase in the NFκB activity score, with a median score of 69.8 (range 54.5–75.0), compared to samples taken at diagnosis (median 38.8, range 27.0–63.8), or samples taken after first-line induction therapy prior to the first ASCT (median 45.7, range 27.0–62.7; Figure 2C). This elevated NFκB activity score remained high after the second ASCT prior to the administration of maintenance/consolidation therapy (median 69.5, range 59.4–77.3). 

By analyzing the longitudinal samples per individual patient, we observed a consistent increase in the NFκB activity score after second-line induction therapy, irrespective of the NFκB activity score at diagnosis or after first-line induction therapy before the first ASCT (Figure 2D). 

Combined, these data show that NFκB pathway activity at diagnosis has no impact on the effectiveness of first-line treatment and can therefore not be used as a predictive marker for therapy response. However, after relapse to first-line total therapy and administration of second-line induction therapy, NFκB pathway activity was significantly increased in the surviving MM PC, and this activity remained high after the second ASCT. This indicates that MM PC with high NFκB pathway activity have a survival advantage over MM PC with low NFκB pathway activity.

### 3.3. BCL2A1 Is the Most Frequently Increased Pro-Survival BCL-2 Member after MM Relapse

Increased expression of pro-survival BCL-2 proteins, of which BCL-XL (*BCL2L1*), BFL-1 (*BCL2A1*), and possibly BCL-2 (*BCL2*) are direct NFκB target genes, is a frequently used mechanism to overcome therapy-induced apoptosis [17]. We therefore analyzed the gene transcript expression of these three BCL-2 family members in samples of the eight MM patients that relapsed after first-line total therapy, as included in dataset GSE19554. Per individual patient, we compared the difference in transcript expression between the sample taken at diagnosis (timepoint 1) and the sample taken after second-line induction therapy (timepoint 3). We observed that in all samples, at least one of the three analyzed transcripts was increased, suggesting that relapse is accompanied by increased expression of pro-survival BCL-2 members (Figure 3A). 

To assess if increased pro-survival BCL-2 expression was specific for members that are direct NFκB target genes, we also analyzed the transcript expression of pro-survival BCL-2 members that are not directly regulated by NFκB: BCL-B (*BCL2L10*), MCL-1 (*MCL1*), and BCL-W (*BCL2L2*). In dataset GSE19554, relapse was associated with >2-fold increased expression of NFκB targets *BCL2A1* in 62.5% of samples, and *BCL2L1* and *BCL2* each in 25% of samples (Figure 3B). For NFκB-independent BCL-2 members a >2-fold increased transcript expression was observed for *BCL2L10* in 25% samples, *MCL1* in 0% of samples, and *BCL2L2* in 12.5% of samples. These results were verified in a second dataset, GSE82307, that contains CD138-purified longitudinal samples of MM patients at diagnosis, and after relapse or progression to first-line total therapy, but before receiving second-line treatment. Additionally, in samples of this second dataset, relapse/progression was more often associated with a >2-fold increased expression of NFκB-mediated pro-survival BCL-2 members than with NFκB-independent pro-survival BCL-2 members. This suggests that MM therapy resistance is associated with increased NFκB pathway activity, and increased expression of pro-survival BCL-2 members that are direct NFκB target genes.

### 3.4. Increased BCL2A1 Expression Correlates with Increased NFκB Target Gene Expression after Relapse

Although mRNA expression encoding for pro-survival BCL-2 family members is heterogeneous, in both datasets with longitudinal samples, GSE19554 and GSE82307, *BCL2A1* (BFL-1) was the most frequently upregulated pro-survival BCL-2 member after MM relapse/progression. Since a role for BFL-1 in MM is still under debate, we tested if differential *BCL2A1* expression after relapse is associated with differential expression of NFκB target genes at this timepoint. In GSE19554, we observed a significant correlation between differential *BCL2A1* expression and the differential expression of 16 NFκB target genes that were incorporated in the STA model (Figure 4A). A total of 8 of these 16 significantly correlating NFκB target genes were also identified in GSE82307, as well as 3 additional NFκB target genes of which differential expression after relapse significantly correlated with differential *BCL2A1*. 

The NFκB molecular cluster in newly diagnosed MM patients included in dataset GSE19784 (Figure 1C) also showed a significantly higher expression of *BCL2A1,* compared to three out of five additional molecular clusters that did not have a significantly increased NFκB activity score (Figure 4B). We observed no such relation between increased NFκB pathway activity and increased *BCL2L1* expression, and even an inverse relation with *BCL2* transcript expression. Since intrinsic apoptosis is not only mediated by pro-survival BCL-2 family members but also by intricate interactions with pro-apoptotic BCL-2 family members, we analyzed the expression of pro-apoptotic effector proteins Bak and Bax [17]. The expression of mRNA encoding for Bak and Bax did not significantly differ between the NFκB molecular clusters and the five additional molecular clusters with significantly lower NFκB pathway activity (Appendix A).

Taken together, these data indicate that increased NFκB target gene expression is correlated with increased *BCL2A1* (BFL-1) expression. In addition to BCL-XL and BCL-2, BFL-1 induction could be a potential mechanism by which NFκB signaling protects MM PC from therapy-induced apoptosis, resulting in disease relapse. Direct or indirect targeting of BFL-1 could therefore provide a novel approach to reduce therapy resistance in a subset of relapsed/refractory (RR)-MM patients.

## 4. Discussion

In this study, we used the novel STA model that allows for the quantification of functional NFκB pathway activity in individual sample gene expression data. We applied this computational model to Affymetrix gene expression microarray datasets containing MM PC samples from diverse patient subsets at multiple stages of disease. Our analysis shows that NFκB pathway activity is very heterogeneous. This is observed in PC from MM patients and is also visible in asymptomatic SMM or pre-malignant MGUS patients, and in PC from healthy donors. By comparing samples from these donor/patient populations, we observed no significant increase in NFκB pathway activity during the early stages of disease development. Although activation of NFκB signaling has been suggested in MGUS-to-MM progression [28], our observations confirm the results of an earlier publication by Annunziata, who showed that an 11-gene NFκB signature was similar in healthy PC, MGUS, and MM samples [4]. These results potentially reflect the dependency on NFκB-activating stimuli from the BM microenvironment that are required for the survival of both healthy PC and early-phase (pre-)malignant MM PC [30].

By stratification of newly diagnosed MM samples based on molecular clusters, we could confirm that the NFκB molecular cluster, which is defined by a high expression of genes involved in the NFκB pathway, also showed the highest STA NFκB pathway activity, with a median score of 44.0, compared to a combined median score of 35.2 in the other molecular clusters. The NFκB molecular cluster is characterized by hyperdiploidy in 66% of cases [2], but NFκB pathway activity in the NFκB molecular cluster was significantly higher compared to NFκB pathway activity in the HY hyperdiploidy molecular cluster, confirming the discriminatory potential of functional NFκB pathway activity scores. In the TC classification by Bergsagel et al., which discriminates eight TC (translocation/cyclin D) groups, the majority of samples in the NFκB molecular cluster are assigned to the D1 group [2]. This D1 group is characterized by overexpression of cyclin D1 compared to healthy PC, without the presence of the five recurrent immunoglobulin translocations that are associated with increased cyclin D expression. Patients within the D1 group showed extensive osteolytic bone disease but were underrepresented in relapsed versus untreated MM and extra-medullary PC leukemia and were suggested to be particularly dependent on BM microenvironment interactions [31]. This favorable prognosis is in accordance with our results, showing that the NFκB activity score is not significantly altered in newly diagnosed MM samples with a poor prognosis, or with a poor response to first-line induction therapy. 

By analyzing longitudinal samples of MM patients undergoing total therapy, we identified a significant increase in NFκB pathway activity in MM PC that survived first-line total therapy, and second-line chemotherapy-based induction therapy after relapse. The median NFκB score in this group was increased by 80% compared to samples taken at diagnosis, and 53% compared to samples taken after first-line induction therapy. This increase in NFκB pathway activity during treatment was observed in samples from all individual MM patients included. Based on our analyses, we cannot conclude if this increase is due to therapy-mediated positive selection of MM PC with high NFκB signaling, or if NFκB signaling is increased in all surviving MM PC due to treatment-induced activation of the BM microenvironment. Data of previous studies are more in accordance with the first clonal selection hypothesis. Mutations resulting in constitutive NFκB pathway activation that render MM PC less dependent on the BM microenvironment for survival are more common in MM cell lines, which often represent more advanced disease, than in primary MM PC [30]. In addition, the PI bortezomib was shown to result in activation, rather than inhibition, of NFκB signaling at late timepoints of exposure in MM PC from cell lines and primary patient samples [32]. 

Irrespective of the mechanism underlying the increased NFκB pathway activity at relapse, MM PC surviving first-line total therapy and second-line induction therapy are likely to be more resistant to therapy-induced cell death. We observed that the increase in NFκB pathway activity during therapy progression was accompanied by increased mRNA expression encoding for at least one member of the pro-survival BCL-2 protein family. Upregulation of these pro-survival BCL-2 transcripts was confirmed but less pronounced in a second dataset containing paired samples of MM patients at diagnosis and after progression/relapse to first-line total therapy. As these patients did not receive second-line therapy before sampling, this may indicate additional therapy-induced clonal selection by increasing selective pressure. In both datasets, increased expression of the direct NFκB target genes (encoding BCL-XL, BFL-1, and BCL-2) was more frequent compared to the NFκB-independent pro-survival BCL-2 members (encoding BCL-B, MCL-1, and BCL-W). Although expression levels were heterogeneous, *BCL2A1* (BFL-1) was most frequently upregulated in samples taken after relapse, and this significantly correlated with the differential expression of NFκB target genes included in the STA NFκB model.

BCL-XL and BCL-2 are known for their contribution to intrinsic apoptosis resistance in MM, and the BCL-2-targeting BH3-mimetic Venetoclax was shown to have clinical efficacy in RR-MM patients harboring a t(11;14) translocation [33]. A potential role for BFL-1 in MM is less clear, since BFL-1 expression is downregulated by transcriptional repressor Blimp-1, which is required for the differentiation of B cells into antibody-secreting PC [19]. Indeed, a study by Tarte et al. showed that *BCL2A1* is strongly repressed in both healthy PC and MM PC purified from patient BM biopsies and cell lines, as compared to peripheral blood and tonsil B cells. In addition, stimulation of an IL-6-dependent MM cell line with NFκB-stimulating cytokines APRIL and BAFF did not result in *BCL2A1* upregulation [34]. On the other hand, two studies by Mitsiades et al. reported contradictory results. Exposure of an MM cell line to the cytokine IGF-1 stimulated NFκB signaling, which was accompanied by upregulation of anti-apoptotic proteins including BFL-1 [35]. In addition, a specific NFκB inhibitor induced apoptosis in MM PC isolated from patients and cell lines. In at least one cell line, this was the result of downregulated NFκB signaling and the subsequent reduction in apoptosis inhibitors including BFL-1 [36]. As MM cell lines are known to harbor more mutations in the NFκB pathway and are able to rapidly proliferate without the presence of a BM microenvironment, these studies further underline the importance of primary patient samples to study potential effects of NFκB-induced BFL-1 expression in MM. 

If BFL-1 expression is indeed a significant mediator of NFκB-induced therapy resistance in MM, (in)direct inhibition of BFL-1 could have therapeutic potential for treatment of RR-MM patients, especially since direct and selective targeting of NFκB signaling upstream of BFL-1 has been proven challenging. The NFκB transcription factor dimers lack accessible hydrophobic pockets that could be exploited for direct inhibition by small molecule inhibitors. Alternative strategies, therefore, are focused on interfering with upstream pathway regulation by IKK and NIK, or even at the receptor signaling level by BCMA and TACI [8]. Despite these efforts, constitutive blocking of NFκB signaling resulted in dose-limiting side effects, including severe infections due to silencing of the immune system [3]. It is hypothesized that inhibiting one of the canonical or non-canonical NFκB pathways is safer, but clinical studies using IKKβ inhibitors, which are expected to specifically block canonical signaling, also showed severe adverse effects [3]. In addition, blocking a signaling pathway can result in unwanted inhibition of the pathway negative feedback loop(s), resulting in increased upstream signaling and additional downstream activation of associated pathways [8]. Blocking the downstream effect of active NFκB signaling by inhibition of BFL-1 could therefore be a preferred option, and this might induce less systemic toxicity compared to general NFκB pathway targeting. Development of BFL-1 inhibitors is still in a pre-clinical phase [37], and additional research is required to prove the efficacy and safety of BFL-1 targeting in MM.

### Future Perspectives

In the current study, we utilized an in silico approach and revealed an increase in NFκB signaling after MM relapse to first-line therapy. In addition, we demonstrated that this was accompanied by increased expression of pro-survival BCL-2 family members BFL-1, BCL-XL, and BCL-2, which are also targets of the NFκB pathway. Future studies should focus on the molecular mechanism to validate the relevance of elevated NFκB signaling and BFL-1 expression in MM relapse, and to demonstrate a potential correlation between these observations. It would be of interest to isolate longitudinal samples of MM patients at diagnosis and after consecutive lines of therapy, in order to be able to analyze NFκB pathway activity using the STA model over a longer disease course. This could be combined with BH3 profiling to determine the level of mitochondrial apoptotic priming, and to analyze the dependence on pro-survival BFL1, BCL-XL, and BCL-2 expression to resist apoptosis [38]. In addition, pharmaceutical inhibition of the NFκB pathway by targeted drugs in relapsed primary MM samples ex vivo could be used to assess potential changes in *BCL2A1*/BFL-1, *BCL2L1*/BCL-XL, and *BCL2*/BCL-2 mRNA and protein expression. 

The STA model for diverse signal transduction pathways is currently being investigated in a multitude of solid tumors and hematological malignancies. Previous publications have already shown that the STA model is a useful tool to discriminate cancer cells from healthy tissue [20,21], and to discriminate cancer patient populations with diverse prognostic outlooks [39]. For multiple signaling pathways, this assay is now adapted for use by quantitative PCR, and commercially available. One of the great advantages of the STA model is that, in addition to being able to identify aberrant pathway activity, these pathways can also be clinically targeted by therapy. This has resulted in the implementation of the STA model in a phase 3 clinical trial for recurrent ovarian carcinoma (NCT03458221). In this clinical study, STA analysis will be performed on histological tumor biopsies, and patients will be treated with targeted drugs to inhibit the predominant pathway. Future studies will have to indicate if the STA model can also be implicated in clinical diagnosis and treatment of MM.

## 5. Conclusions

In this study, we quantified functional NFκB pathway activity in healthy PC or (pre-)malignant MM PC from specific patient subgroups at various stages of disease using the computational STA model. We found that the NFκB pathway activity was higher in MM PC from newly diagnosed patients in the NFκB cluster compared to the other molecular clusters but observed no additional differences in NFκB pathway activity in relation to early disease development, patient prognosis, or response to first-line therapy. However, MM PC that survived first-line total therapy showed significantly increased NFκB pathway activity at relapse, compared to the MM PC from samples taken in the pre-treatment phase. In a subset of relapsed samples, this increase in NFκB pathway activity was accompanied by an increased expression of the *BCL2A1* transcript encoding pro-survival BFL-1. We hypothesize that upregulation of BFL-1, in addition to NFκB targets BCL-XL and BCL-2, could contribute to the therapy resistance of MM PC and propose that (in)direct targeting of BFL-1 may provide a new approach for a subset of RR-MM patients.

## Figures and Tables

**Figure 1 cancers-13-04668-f001:**
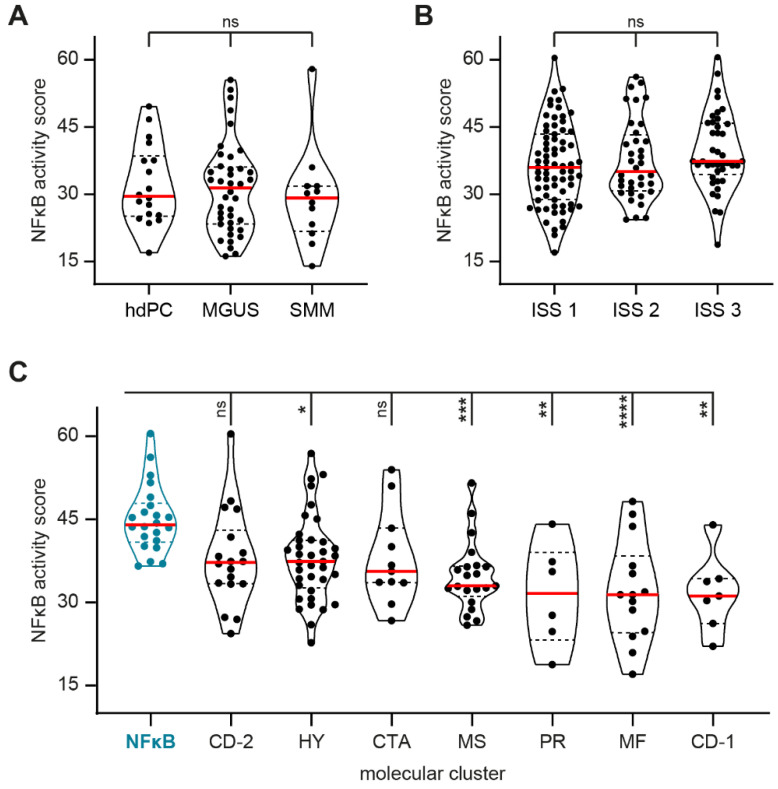
Violin plots showing NFκB activity score as analyzed by signal transduction pathway activity (STA) analysis in plasma cells (PC) from (**A**) healthy donors (hdPC), pre-malignant monoclonal gammopathy of undetermined significance (MGUS) patients, and asymptomatic smoldering multiple myeloma (SMM) patients, as incorporated in dataset GSE5900; (**B**) newly diagnosed MM patients, stratified by the international staging system (ISS) into good-prognosis ISS 1, intermediate-prognosis ISS 2, and poor-prognosis ISS 3, as incorporated in dataset GSE19784; (**C**) newly diagnosed MM patients, stratified by molecular clusters, as incorporated in dataset GSE19784. Included samples are represented by individual data points, solid red lines indicate the median, and dashed lines indicate the quartiles of the population. ns, not significant, * *p* < 0.05, ** *p* < 0.01, *** *p* < 0.001, **** *p* < 0.0001.

**Figure 2 cancers-13-04668-f002:**
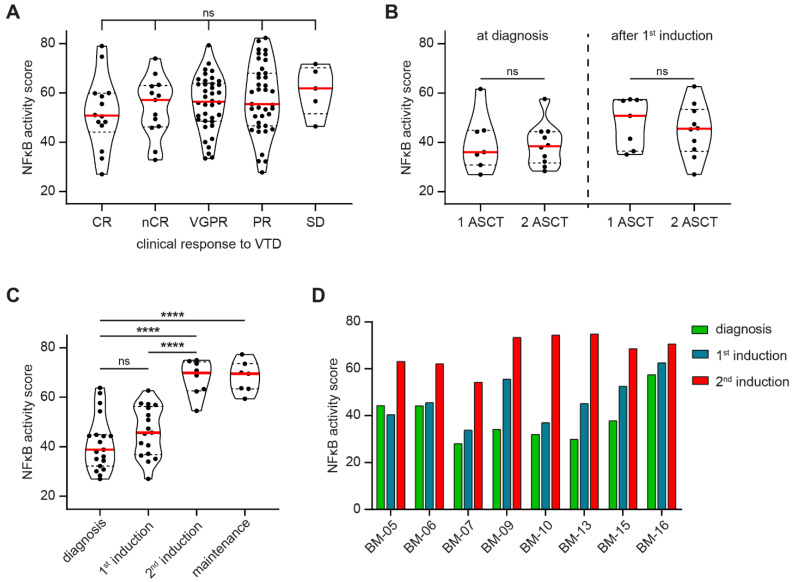
Violin plots showing the NFκB activity score as analyzed by STA analysis in PC from (**A**) newly diagnosed MM patients with a quantified clinical response to subsequently administered first-line VTD (dexamethasone-thalidomide-bortezomib) induction therapy: complete response (CR), near-complete response (nCR), very good partial response (VGPR), partial response (PR), and stable disease (SD), as incorporated in dataset GSE68871; (**B**) MM patients at diagnosis, and after first-line chemotherapy-based induction therapy, that were stratified into two groups based on clinical response to subsequent total therapy: patients that required 1 ASCT, and patients that relapsed after the first ASCT and required a second ASCT (2 ASCT), as incorporated in dataset GSE19554; (**C**) MM patients at diagnosis, after first-line chemotherapy-based induction therapy prior to the first ASCT (1st induction), after second-line chemotherapy-based induction therapy prior to the second ASCT (2nd induction), and after the second ASCT before the start of maintenance/consolidation therapy (maintenance), as incorporated in dataset GSE19554. Included samples are represented by individual data points, solid red lines indicate the median, and dashed lines indicate the quartiles of the population. ns, not significant, **** *p* < 0.0001. (**D**) Bar graph showing NFκB activity score as analyzed by STA analysis in longitudinal PC samples of 8 individual MM patients at diagnosis, after first-line chemotherapy-based induction therapy prior to the first ASCT (1st induction), and after second-line chemotherapy-based induction therapy prior to the second ASCT (2nd induction), as incorporated in dataset GSE19554. BM, bone marrow.

**Figure 3 cancers-13-04668-f003:**
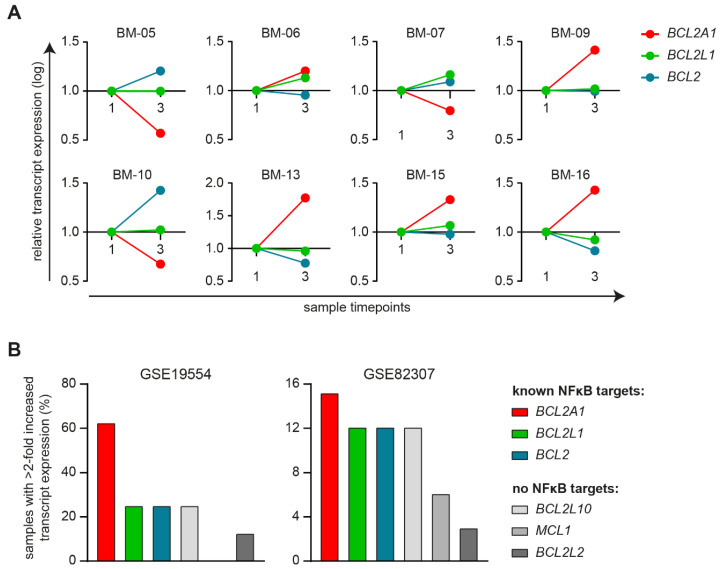
(**A**) Plots showing relative mRNA transcript expression of *BCL2A1* (BFL-1), *BCL2L1* (BCL-XL), and *BCL2* (BCL-2), in PC taken at diagnosis (timepoint 1), and after second-line chemotherapy-based induction therapy after relapse to first-line total therapy (timepoint 3), from 8 individual MM patients, as included in dataset GSE19554. (**B**) Bar graphs showing the percentage of MM PC patient samples that showed >2-fold increased mRNA transcript expression of NFκB-regulated pro-survival targets *BCL2A1*, *BCL2L1*, and *BCL2*, and NFκB-independent pro-survival members *BCL2L10* (encoding BCL-B), *MCL1* (encoding MCL-1), and *BCL2L2* (encoding BCL-W), after relapse to first-line total therapy, with (dataset GSE19554) or without (GSE82307) subsequent administration of second-line induction therapy, in comparison to mRNA transcript expression at diagnosis.

**Figure 4 cancers-13-04668-f004:**
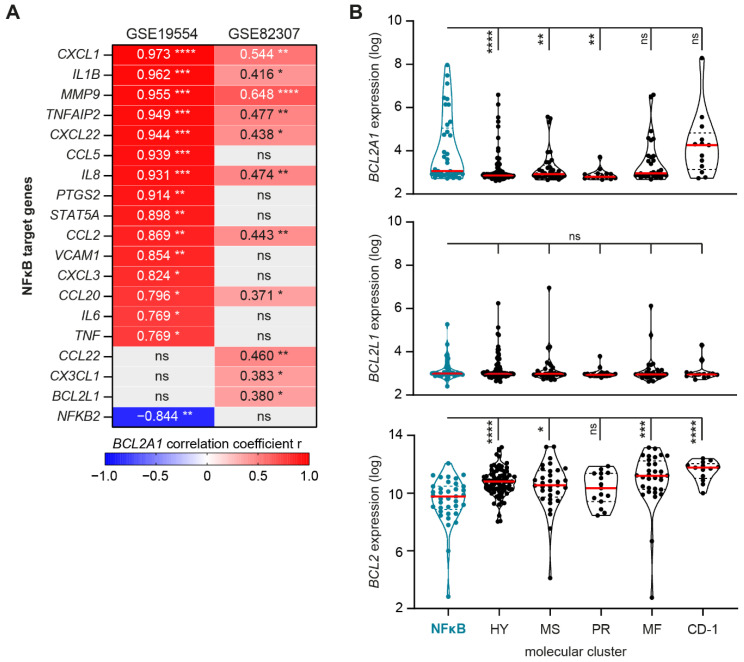
(**A**) Heatmap showing correlation coefficient (R value) between differential *BCL2A1* expression and differential expression of NFκB target genes included in the STA analysis, as determined by mRNA transcript expression in PC from MM patients at diagnosis versus mRNA transcript expression after relapse to first-line total therapy, with (dataset GSE19554) or without (GSE82307) subsequent administration of second-line induction therapy. (**B**) Violin plots showing *BCL2A1*, *BCL2L1*, and *BCL2* mRNA transcript expression in PC of newly diagnosed MM patients, as incorporated in dataset GSE19784, in molecular cluster NFκB, and in molecular clusters HY, MS, PR, MF, and CD-1 that all showed a significant lower STA NFκB activity score compared to molecular cluster NFκB in Figure 1C. Included samples are represented by individual data points, solid red lines indicate the median, and dashed lines indicate the quartiles of the population. ns, not significant, * *p* < 0.05, ** *p* < 0.01, *** *p* < 0.001, **** *p* < 0.0001.

**Table 1 cancers-13-04668-t001:** Samples included in STA NFκB analysis per dataset.

Dataset	Reference	Description	Included Samples
GSE5900	Zhan et al. [22]	Untreated samples from healthy donors, MGUS, and SMM patients	hd PC *n* = 18
			MGUS *n* = 40
			SMM *n* = 12
GSE19784	Broyl et al. [2]	Newly diagnosed MM with specified molecular clusters ^1^ Newly diagnosed MM with specified ISS stage defining clinical prognosis	ISS 1 *n* = 71
			ISS 2 *n* = 36
			ISS 3 *n* = 41
			CD-1 *n* = 7
			CD-2 *n* = 18
			MF *n* = 14
			MS *n* = 21
			PR *n* = 6
			HY *n* = 37
			NFκB *n* = 22
			CTA *n* = 11
GSE68871	Terragna et al. [23]	Newly diagnosed MM with clinical response to subsequent first-line VTD induction therapy	CR *n* = 14
			nCR *n* = 13
			VGPR *n* = 38
			PR *n* = 39
			SD *n* = 5
GSE19554	Zhou et al. [24]	Longitudinal MM samples during first-line and second-line total therapy including ASCT	diagnosis *n* = 19
			1st-line induction *n* = 17
			2nd-line induction *n* = 8
			maintenance *n* = 7
GSE82307	Weinhold et al. [25]	Matched MM samples at diagnosis and after progression/relapse to first-line total therapy ^2^	diagnosis *n* = 33
			relapse *n* = 33

^1^ The SOCS3/PRL3 cluster contained only 3 samples and was therefore not included. The previously reported contaminated myeloid cluster was excluded from further analysis. ^2^ Samples were not analyzed by STA but on individual probeset intensity. MGUS, monoclonal gammopathy of undetermined significance; SMM, smoldering multiple myeloma; hd, healthy donor; MM multiple myeloma; ISS, international staging system; VTD bortezomib-thalidomide-dexamethasone; CR, complete response; nCR, near-complete response; VGPR, very good partial response; PR, partial response; SD, stable disease; ASCT autologous stem cell transplant.

## Data Availability

Publicly available datasets were analyzed in this study. These data can be found at the Gene Expression Omnibus (https://www.ncbi.nlm.nih.gov/geo/ (accessed on 18 February 2020)) under accession numbers: GSE5900, GSE19784, GSE68871, GSE19554, and GSE82307.

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
