# Peer review of "Multiple Myeloma Relapse Is Associated with Increased NFκB Pathway Activity and Upregulation of the Pro-Survival BCL-2 Protein BFL-1"

_cancers, 2021, doi:10.3390/cancers13184668_

Round 1

Reviewer 1 Report

This is a well-written paper with rigorous methods that shows how NFkB expression is upregulated in MM patients who relapse after 1st line therapy. The authors also show that expression of BFL-1, a pro-survival protein downstream of NFkB, is elevated in relapsed patients. The authors conclude that the increased expression of BFL-1 is likely due to NFkB and is likely mediating increased survival of MM cells leading to relapse. I think the conclusion regarding the connection between BFL-1 upregulation and NFkB upregulation/relapse could be better supported as it seems to be a very heterogeneous response (Fig. 3A) and it looks like it may be caused more by selection caused by 2nd line therapy than the relapse itself (Fig. 3B and Fig. 4A). I think that the connection between NFkB/relapse and BFL-1 in MM could be more clearly presented/explained, especially given the controversy of the topic and contradictory reports in the literature (BFL-1 is reported to be downregulated in healthy plasma cells and MM cells, Ref: Tarte et al., 2004). The authors can better explain their argument or blunt their conclusions to address this criticism. This study would also benefit from in vitro and/or in vivo mechanistic studies that could better prove the arguments being made by the authors and better illustrate the relevance of NFkB/BFL-1 in relapse. I think that this limitation should be pointed out by the authors and potential follow-up studies should be proposed in the discussion/conclusions. 

Thank you for your hard work on this interesting, well-constructed manuscript. 

Author Response

This is a well-written paper with rigorous methods that shows how NFkB expression is upregulated in MM patients who relapse after 1st line therapy. The authors also show that expression of BFL-1, a pro-survival protein downstream of NFkB, is elevated in relapsed patients. The authors conclude that the increased expression of BFL-1 is likely due to NFkB and is likely mediating increased survival of MM cells leading to relapse.

--> We thank the reviewer for appreciating our manuscript.

I think the conclusion regarding the connection between BFL-1 upregulation and NFkB upregulation/relapse could be better supported as it seems to be a very heterogeneous response (Fig. 3A) and it looks like it may be caused more by selection caused by 2nd line therapy than the relapse itself (Fig. 3B and Fig. 4A). I think that the connection between NFkB/relapse and BFL-1 in MM could be more clearly presented/explained, especially given the controversy of the topic and contradictory reports in the literature (BFL-1 is reported to be downregulated in healthy plasma cells and MM cells, Ref: Tarte et al., 2004). The authors can better explain their argument or blunt their conclusions to address this criticism.

--> We thank the reviewer for this comment. Although our analyses of primary MM samples at diagnosis and after relapse to first-line treatment show increased NFκB activity, an increase in BCL2A1 expression in a subset of patients, and a correlation between BFL-1 expression and NFκB target gene expression, these studies are indeed not sufficient to prove a direct correlation between these observations. Throughout our manuscript (visible by track-changes) we now put an extra emphasis on the heterogeneous expression of BFL-1 (lines 307 and 407), and the importance of BCL-XL and BCL-2 (lines 36, 330, and 488). In the discussion we suggest that second-line induction therapy (as in samples from dataset GSE19554) may provide additional therapy-induced clonal selection, compared to the samples of patients that only received first-line therapy (dataset GSE82307; lines 400-404). Furthermore, in the first paragraph of novel section 4.1 “future perspectives” we now propose future mechanistic studies to address these points of concern. We removed claims of “NFκB-mediated BFL-1 upregulation” from the manuscript (lines 36 and 488).

This study would also benefit from in vitro and/or in vivo mechanistic studies that could better prove the arguments being made by the authors and better illustrate the relevance of NFkB/BFL-1 in relapse. I think that this limitation should be pointed out by the authors and potential follow-up studies should be proposed in the discussion/conclusions.

--> We thank the reviewer for the opportunity to elaborate on potential follow-up studies. We now included the limitation of an in silico approach, and provide suggestions for follow-up mechanistic studies in the first paragraph of novel section 4.1 “future perspectives”.

Reviewer 2 Report

In this Manuscript by Spaan et al, a signal transduction pathway activity model is employed to quantify NFκB pathway activity in multiple myeloma plasma cells from patient datasets. The authors found that NFκB pathway activity does not change in several contexts, but is increased in disease relapse after treatment. This correlates with the expression of BCL2A1, a member of the BCL2 family of proteins.

The study is interesting, specially in light of recent developments in the field of Bcl2 proteins and personalised medicine.

I have only minor comments related to this work:

Does this STA signature correlates or change with other clinical stages or clinical features of MM?

In figure 4B mRNA expression levels of several anti-apoptotic proteins (in different molecular cluster) are showed. As intrinsic apoptosis is the results of an intricate interaction network among several proteins, it would be interesting see also the levels of pro-apoptotic proteins such as Bax and Bak.

Do the authors think that the STA model has some sort of translational possibility in the clinical practice? I think this should be discussed.

Author Response

In this Manuscript by Spaan et al, a signal transduction pathway activity model is employed to quantify NFκB pathway activity in multiple myeloma plasma cells from patient datasets. The authors found that NFκB pathway activity does not change in several contexts, but is increased in disease relapse after treatment. This correlates with the expression of BCL2A1, a member of the BCL2 family of proteins. The study is interesting, specially in light of recent developments in the field of Bcl2 proteins and personalised medicine.

--> We thank the reviewer for appreciating the relevance of our work.

I have only minor comments related to this work:

Does this STA signature correlates or change with other clinical stages or clinical features of MM?

--> We thank the reviewer for this question. We recognize the importance to study the relation between the NFκB STA signature and clinical stages and clinical features of MM. In our manuscript we analyzed the STA signature in the context of early disease development  (healthy donor plasma cells, MGUS, and smoldering MM; Figure 1A), newly diagnosed MM in different ISS stadia (Figure 1B), and during first-line treatment therapy (Figure 2A-B), and observed no statistically significant changes. It would certainly be interesting to analyze datasets with patient samples representing later disease stages, including extra-medullary MM and/or plasma cell leukemia (PCL). This is of particular interest as the malignant cells become less dependent on the bone marrow micro-environment for survival during these stages, and this might impact NFκB pathway activity. Unfortunately, datasets containing extra-medullar MM and/or PCL samples are, to the best of our knowledge, not available for the current platform.

In addition to the disease stages and cytogenetics that we analyzed in our manuscript, it would also be interesting to study the relation between the STA signature and clinical features, as defined by the CRAB criteria. Although the available datasets that we analyzed do not specify any additional clinical information per sample, some clinical features are associated with identified MM molecular clusters. As described in the discussion of our manuscript, the majority of samples from the NFκB molecular cluster, which also showed a significant increased NFκB STA score, are assigned to the D1 group of the TC classification (Bergsagel et al., Blood 2005). This D1 group is associated with extensive osteolytic bone disease. Furthermore, the samples of molecular clusters PR, MS, and MF have previously been identified to have statistically significant higher levels of serum B2M and lower levels of serum albumin, compared to other molecular clusters (Zhan et al., Blood 2006). We therefore compared the STA signature in all MM molecular clusters that were incorporated in the analysis (new Supplemental Figure 1). Interestingly, we observed no statistically significant changes in the STA score when we compared all molecular clusters, except for the NFκB cluster. All data combined, this validates the specificity of the STA model and highlights the importance of measuring STA pathway activity in addition to clinical features that are measured as standard-of-care.

In figure 4B mRNA expression levels of several anti-apoptotic proteins (in different molecular cluster) are showed. As intrinsic apoptosis is the results of an intricate interaction network among several proteins, it would be interesting see also the levels of pro-apoptotic proteins such as Bax and Bak.

--> We thank the reviewer for this comment. In the original manuscript we showed a significant increase in pro-survival BFL2A1/BFL-1 expression in the NFκB molecular cluster (Figure 4B), compared to the molecular clusters of which we previously demonstrated to have a statistically lower NFκB STA score (Figure 1C). This difference was not observed for pro-survival BCL2L1/BCL-XL expression, and pro-survival BCL2 expression was even increased in molecular clusters with lower NFκB STA scores (Figure 4B). As the reviewer suggests, we now also analyzed expression of pro-apoptotic Bak and Bax in these molecular clusters (new Supplementary Figure 2). We observed no statistical differences in BAK and BAX expression between molecular clusters with significant different NFκB STA scores. This indicates that upregulation of pro-survival BFL2A1/BFL-1 can increase the resistance to apoptosis of MM PC with high NFκB activity.

Do the authors think that the STA model has some sort of translational possibility in the clinical practice? I think this should be discussed.

--> We thank the reviewer for the opportunity to elaborate on the translational opportunities of the STA model in the clinic. We indeed expect implementation of the STA model into clinical practice, and this discussion is now added to the second paragraph of novel section 4.1 “future perspectives”.